# Integrating Multi-Omics in Endometrial Cancer: From Molecular Insights to Clinical Applications

**DOI:** 10.3390/cells14171404

**Published:** 2025-09-08

**Authors:** Hye Kyeong Kim, Taejin Kim

**Affiliations:** 1CHA University Fertility Center Ilsan, Goyang-si 10414, Republic of Korea; hazlendhye1@chamc.co.kr; 2Department of Urology, CHA University Ilsan Medical Center, CHA University School of Medicine, Goyang-si 10414, Republic of Korea

**Keywords:** endometrial cancer, multi-omics, molecular classification, genomics, metabolomics, precision medicine, The Cancer Genome Atlas (TCGA)

## Abstract

Endometrial cancer (EC) is the most common gynecologic malignancy in developed countries, and its incidence is increasing globally. While early-stage ECs generally show good prognosis, advanced or recurrent cases and those with aggressive histologic subtypes exhibit poor outcomes. Traditional histopathologic classification, however, fails to reflect the molecular heterogeneity of EC, limiting its role in guiding treatment. Recent developments in multi-omics have enhanced our understanding of EC biology, which supports more personalized treatment strategies. The Cancer Genome Atlas (TCGA) classification has provided a more systematic molecular framework for stratifying risk and identifying prognostic and therapeutic biomarkers. This review discusses the latest developments in multi-omics-based classification of EC, highlights emerging diagnostic and therapeutic strategies, and summarizes ongoing clinical trials that aim to translate molecular discoveries into improved outcomes.

## 1. Introduction

Endometrial cancer (EC), a malignancy arising from the uterine endometrium, is the most prevalent gynecologic cancer in developed nations [1] and shows a growing incidence globally. While cervical cancer incidence in developing countries is decreasing due to human papillomavirus vaccination and routine screening [2,3], westernized lifestyle, population aging, and declining fertility [4] have led to a 132% increase in the incidence of EC over the last 30 years [1].

Traditionally, EC has been diagnosed based on histopathological features [5]. This method is effective for staging and guiding treatment strategy. However, this approach falls short in capturing biological heterogeneity of EC. Although early-stage ECs typically show favorable prognosis with a 5-year survival rate of nearly 95% [1,6], advanced or recurrent disease, as well as ECs with aggressive histologic subtypes, are associated with poor outcomes [5]. In these cases, the five-year survival rate remains low, and conventional treatments such as chemotherapy and radiotherapy frequently show limited efficacy [1,6].

One major limitation of the current treatment paradigm stems from the conventional classification system of EC. This system fails to adequately reflect the molecular complexity of the disease. This has led to the development of next-generation classification strategies incorporating multi-omics such as genomics, transcriptomics, proteomics, and metabolomics. Multi-omics provides a more accurate understanding of tumor pathophysiology, facilitating personalized diagnostic and therapeutic approaches [7,8].

Typically, ECs are driven by the interplay of hormonal factors, genetic and molecular aberrations [9]. Development of estrogen-dependent endometrial glandular hyperplasia to invasive carcinoma is orchestrated by hormonal influences and dynamic glandular–stromal interactions. In normal endometrium, estrogen induces endometrial proliferation and progesterone, a counteracting hormone, sheds the endometrium. Prolonged unopposed estrogen exposure to endometrium, however, causes excessive proliferation of glandular epithelium and initiates hyperplastic growth. The deficiency of progesterone accelerates endometrial hyperplastic lesion toward malignant change [10]. Obesity, polycystic ovarian syndrome, early menarche, late menopause, nulliparity, and the long term use of tamoxifen can contribute to the excessive estrogen exposure in the endometrium [1,6,11,12,13,14].

Beyond the well-known hormonal and genomic drivers, stromal and immune microenvironmental changes also contribute to the progression of ECs [6]. Endometrial stroma undergoes profound remodeling process during its switch from a precancerous EIN (endometrial intraepithelial neoplasia) lesion to invasive carcinoma. Quiescent fibroblasts are activated into cancer-associated fibroblasts (CAFs) with a myofibroblastic phenotype [15]. These CAFs modulate a desmoplastic stromal response, depositing and remodeling extracellular matrix (ECM) components and secreting matrix metalloproteinases (MMPs) that degrade basement membranes [16]. This environment provides a favorable situation for invasion and dissemination. High levels of gelatinases MMP2 and MMP9 show correlation with advancing FIGO stage and poorer prognosis in EC [17]. Concurrently, immune dysregulation plays a significant role in cancer progression. Tumor-associated macrophages (TAMs) within the EC microenvironment tend to polarize toward the M2 phenotype, secreting immunosuppressive cytokines such as IL-10 and TGF-β that blunt CD8^+^ T-cell activity, as well as pro-angiogenic factors like VEGF that facilitate neovascularization [18]. Studies have revealed that high densities of M2-polarized TAMs are strongly associated with higher tumor grade, advanced stage, and reduced survival. The tumor–stromal crosstalk further promotes immune evasion through checkpoint pathways and inflammatory signaling [19]. Recent evidences acquired from transcriptomic and proteomic studies have begun to reveal how these interactions are organized in situ. For example, integrated single-cell and spatial transcriptomics in EC has identified distinct paracrine signaling circuits. According to the authors, midkine (MDK) produced by carcinoma cells can engage nucleolin (NCL) receptors on adjacent stromal/endothelial cells. It effectively educates the local stroma and promotes immune exclusion. Spatial profiling confirms that such MDK–NCL signaling occurs between neighboring cells within the tumor tissue and that regions with active NCL signaling coincide with lower immune cell infiltration [20]. Altogether, these findings underscore that the pathogenesis of endometrial cancer is not a purely epithelial-cell-autonomous process but a dynamic, multi-cellular evolution, in which stromal remodeling and immune microenvironment modulation facilitate the transition from EIN to carcinoma and sustain tumor growth.

The role of multi-omics in ECs became more clear in the last decade [8,21]. Exome sequencing studies showed the common mutations such as PTEN, PIK3CA, ARID1A and TP53, and these findings were the basis for TCGA molecular subtypes [22]. Transcriptomics, the analysis of RNA expression from tumor tissue, has provided more information about gene expression patterns, immune signatures and non-coding RNAs which can be related to treatment response [23,24,25]. Proteomics, which is the large scale study of protein expression and modification, has demonstrated the real protein changes and pathway activation in cancer [23]. Lastly, metabolomics investigates metabolites and metabolic pathways in diseases. This approach revealed the altered energy metabolism and metabolites in ECs. It is a potential biomarker source [26]. Integrated data of exome, transcriptome, proteome and metabolome finally changed the classification of EC from morphology based to molecular based system [22,27]. This multi-omics approach also helps to develop prognostic models and to develop targeted therapy and immunotherapy [8,21].

Several biomarkers were reported from multi-omics studies in EC. The most frequent mutations are PTEN, PIK3CA, TP53 and ARID1A, which are important in the development of EC [22,23]. POLE mutation and microsatellite instability (MSI) are strong prognostic markers, with POLE mutation showing good prognosis and MSI is linked with Lynch syndrome [27]. CTNNB1 and KRAS mutations also have clinical meaning in tumor progression [22,28,29]. On protein level, abnormal activation of PI3K/AKT/mTOR and WNT/β-catenin pathways has been confirmed [23], and some circulating proteins like annexin A2 and heat shock proteins are suggested as potential biomarkers [30,31,32]. These markers may help with diagnosis, prognosis and treatment decisions in EC [22,23,27]. A summary of important genes and proteins is presented in Table 1.

In this review, we discuss recent advances in the molecular classification of EC and their clinical application. We discuss how multi-omics-driven insights are reshaping diagnostic algorithms and therapeutic strategies. Finally, we summarize ongoing clinical trials to provide a perspective on future directions and the potential integration of multi-omics into routine clinical practice.

## 2. Classification of Endometrial Cancer

### 2.1. Histologic Classification of Endometrial Cancer

Endometrial cancer has traditionally been classified based on endocrine features and histopathologic characteristics. In 1983, Bokhman et al. classified EC into estrogen-dependent (type I) and estrogen-independent (type II) [33]. Type I is mainly composed of endometrioid carcinoma and is associated with a relatively favorable prognosis, while type II includes aggressive subtypes such as serous and clear cell carcinoma.

This dualistic model was later broadened by the World Health Organization (WHO) and the International Federation of Gynecology and Obstetrics (FIGO). Over time, WHO and FIGO have developed histologic and staging systems to improve the diagnosis, prognosis, and treatment of endometrial cancer. The WHO classification of EC provides a standardized guideline based on microscopic morphology and cellular features. According to this system, EC is subdivided into endometrioid, serous, clear cell, mucinous, squamous cell, transitional cell, small cell, and undifferentiated, based on histologic characteristics [34]. The staging of EC has been largely dependent on nuclear grade and glandular architecture. For clarity, Table 2 provides a simplified comparison of the major classification systems, outlining their criteria, representative subtypes, and prognostic relevance (Table 2).

Meanwhile, advances in molecular and genetic technologies have provided substantial and meaningful insights into the nature of EC. This led to the introduction of The Cancer Genome Atlas (TCGA). Reflecting the TCGA framework, WHO and FIGO released updated staging systems in 2020 and 2023, respectively (Appendix A, Table A1) [1,5]. These aim to better define prognostic groups and support more individualized treatment strategies (Appendix A, Table A2). An integrated version of FIGO staging with molecular classification is provided in Appendix A (Table A1 and Table A2) [5].

This classification system based on histology, however, presents several limitations. First, there is often an overlap between histologic subtypes and grade determination. For instance, it is difficult to clearly distinguish high-grade endometrioid carcinoma from serous carcinoma because of their histologic similarities, even though this pathologic diagnosis is critical for treatment planning. Molecular determinants can supplement traditional immunochemical staining and thus enable more accurate diagnosis. Furthermore, there are interpersonal and inter-institutional disparities in the conventional classification system. Implementing molecular classification may help reduce such bias. This shift laid the foundation for a novel molecular taxonomy, as first established by The Cancer Genome Atlas (TCGA) [22].

### 2.2. TCGA-Based Four Molecular Subtypes

In 2013, TCGA published an integrated genomic, transcriptomic, and proteomic analysis of hundreds of endometrial tumors, revealing that EC is not a single disease but rather comprises four distinct molecular subtypes [22]. This molecular classification has profound prognostic implications and has paved the way for a new paradigm in the management of EC. The four subtypes are as follows (Figure 1):

#### 2.2.1. POLE Ultramutated (POLEmut)

POLE is a gene encoding catalytic subunit of DNA polymerase epsilon participating in nuclear DNA replication and repair [22,36]. POLE ultramutated accounts for approximately 5–15% of ECs [37]. ECs with POLEmut exhibit an extremely high mutation rate with high tumor mutational burden but paradoxically show the most favorable prognosis among all subtypes, regardless of histological grade [22].

Surrogate methods for detecting POLEmut include next-generation sequencing (NGS), Sanger sequencing, and targeted hot-spot panels focusing on the exonuclease domain. To classify an EC as POLEmut, the presence of at least one of pathogenic mutations in POLE gene is mandatory. These mutations include P286R, V411L, S297F, S459F, A456P, F367S, L424I, M295R, P436R, M444K, and D368Y. Variants of uncertain significance (VUS) or mutations outside this definition should not be used to assign POLEmut status [7,37].

Histologically, POLEmut ECs are strongly associated with high-grade EECs with prominent lymphocytic infiltration, marked nuclear atypia, and abundant tumor-infiltrating lymphocytes. The strong immune response may partly explain the favorable outcome of this subtype [38]. Current guidelines recommend de-escalation of cancer staging and omission of adjuvant therapy in this group, even in the presence of other high risk features [5].

#### 2.2.2. Mismatch Repair Deficient (MMRd)/Microsatellite Instability-High (MSI-H)

Comprising about 25–30% of ECs, this subtype is defined by the loss of function in one or more mismatch repair (MMR) proteins, most commonly MLH1, PMS2, MSH2, and MSH6 [37]. Surrogate markers for detecting MMR deficiency include immunohistochemical staining for these four MMR proteins. Loss of expression of any of these proteins can be diagnosed as MMRd. Specifically, loss of MLH1 is often caused by epigenetic silencing through promotor hypermethylation, while the loss of MSH2 or MSH6 may indicate a germline mutation associated with Lynch syndrome. Therefore, patients with MMRd ECs ought to undergo germline genetic counseling and testing for hereditary cancer syndromes [37].

MMRd ECs are frequently associated with high-grade endometrioid histology, and their prognosis is intermediate. Notably, this subtype shows a significant response to immune checkpoint inhibitors, especially those targeting programmed cell death protein 1 (PD-1)/programmed death-ligand 1 (PD-L1). It is attributed to their high neoantigen load and immunogenic microenvironment [39]. KEYNOTE-158 and the GARNET study have reported favorable response rates and durable progression-free survival in patients with advanced or recurrent MMRd ECs treated with anti–PD-1 agents such as pembrolizumab or dostarlimab [39,40,41]. Therefore, MMR status serves not only as a molecular classifier but also as a predictive biomarker for immunotherapy eligibility.

#### 2.2.3. Copy-Number High (CN-H)/p53-Abnormal (p53abn)

Copy-number high, also known as “p53abn”, represents approximately 15–20% of ECs [37]. This subtype displays extensive somatic copy-number alterations and frequent TP53 gene mutations. Most serous carcinomas, an aggressive histologic type, and a part of high-grade EECs belong to this group [22,42].

p53abn ECs generally lack hormone receptor expression, exhibit substantial chromosomal instability, and rarely display microsatellite instability [22]. Clinically, they represent the poorest prognostic group among the four TCGA molecular classes, often presenting with early relapse, lymphovascular space invasion (LVSI), distant spread, and cancer-related mortality [37]. Among the four molecular subtypes, the CN-H/p53abn group exhibits the most unfavorable prognosis and is strongly linked to increased recurrence rates and cancer-related mortality. Adjuvant therapy is routinely recommended in this group, including chemotherapy and/or radiotherapy [43]. Efforts to enhance treatment efficacy have recently included the investigation of targeted therapies. In particular, human epidermal growth factor receptor (HER2) amplification, frequently identified in serous endometrial carcinoma, has emerged as a potential therapeutic marker. A key trial showed that adding trastuzumab—a monoclonal antibody targeting HER2—to standard chemotherapy improved progression-free survival [44]. In addition, immune checkpoint inhibitors have shown activity, and novel agents such as antibody-drug conjugates and inhibitors targeting molecules like Wee1 and cyclin E1 (CCNE1) are currently under investigation in this high-risk cohort [45,46].

In the CN-H subtype, further refinement of prognostic and therapeutic strategies can be achieved by integrating co-mutation profiles—such as ARID1A loss or CCNE1 amplification—or transcriptional signatures [47].

#### 2.2.4. Copy-Number Low (CN-L)/No Specific Molecular Profile (NSMP)

Among the four molecular subtypes, tumors classified as having no specific molecular profile (NSMP), also termed CN-L, are the most prevalent group, comprising roughly 30–60% of ECs [37,43,48]. Tumors that do not meet the defining features of the other TCGA molecular categories are defined as NSMP. To be specific, these tumors lack pathogenic POLE mutations, exhibit normal p53 expression, and maintain intact MMR functionality.

This subtype is commonly associated with low-grade, estrogen receptor–positive endometrioid endometrial carcinomas (EECs), and it is generally linked to an intermediate clinical outcome. Recent studies have proposed that additional biomarkers—such as CTNNB1 mutations within exon 3 [49], elevated expression of L1 cell adhesion molecule (L1CAM) [50], and progesterone receptor (PR) loss [51]—could be useful for further assessing recurrence risk and tailoring treatment within this heterogeneous group [51].

Although traditionally considered a lower-risk category, NSMP tumors with CTNNB1 alterations have been connected to higher recurrence rates, particularly in early-stage disease and among younger patients [49]. Current research is focused on subdividing this group more precisely using integrated transcriptomic and epigenomic analyses, with the goal of enhancing prognostic accuracy and guiding individualized therapeutic decisions [28].

### 2.3. Prognostic and Therapeutic Implications

Molecular subclassification has increasingly been incorporated into current risk assessment criteria, including the 2021 ESGO/ESTRO/ESP guidelines [27] and the updated 2023 FIGO staging system [5]. These molecular features now support clinical decision-making, particularly in tailoring treatment intensity to individual tumor profiles. For example, ECs with POLEmut alterations are generally associated with favorable outcomes, which has led to a growing trend toward minimizing adjuvant therapy in these cases [27]. In contrast, tumors classified as p53abn group often demonstrate aggressive behavior and poor prognosis, typically requiring more intensive adjuvant strategies. Similarly, in patients with advanced or recurrent MMRd tumors, immunotherapy using pembrolizumab and dostarlimab has yielded promising clinical responses, reflecting the high immunogenicity of this subgroup [52].

As molecular profiling continues to expand our understanding of tumor biology, it is expected to enhance prognostic modeling and support the development of individualized treatment algorithms tailored to each molecular subtype [5,27]. Several studies also compared the TCGA molecular classification with the classical histology and ER status system, consistently showing stronger associations with outcomes for the molecular groups [48,53,54]. For example, in PORTEC-3, POLE-ultramutated tumors had excellent survival, whereas p53abn cancers had the worst outcomes, a separation that histology alone could not account for [54]. In some reports, TCGA/ProMisE schemes improved risk stratification and predicted treatment benefit, supporting their integration into clinical decision pathways [54,55]. Although the exact values vary among studies, the overall evidence supports that TCGA molecular classification provides more accurate risk stratification for EC patients. The prognostic separation of these four molecular groups is illustrated in Figure 2, showing excellent outcomes in POLE ultramutated tumors, poor survival in p53abn cancers, and intermediate outcomes in MSI-hypermutated and CN-L groups [22,54].

## 3. Key Signaling Pathways and Therapeutic Targets Uncovered by Multi-Omics

Recent progress in multi-omics approaches has deepened our insight into the molecular mechanisms causing cancer [8,21]. Pathways involved in the tumor initiation, progression, and therapeutic response have emerged as of principal interest. These insights also suggest several promising targets for precision medicine. In this section, we describe the most relevant signaling pathways identified through integrative genomic, transcriptomic, and proteomic studies, and discuss their potential clinical implications [23]. Before detailing individual signaling pathways, it is important to highlight the overall genomic landscape of EC. Figure 3 summarizes the distribution and co-occurrence of recurrent alterations across the four TCGA-defined molecular subtypes, as illustrated in the TCGA-UCEC cohort [22]. This Oncoprint view shows the high prevalence of PTEN, PIK3CA, ARID1A, CTNNB1, and TP53 mutations, and provides a framework for understanding how these alterations drive tumorigenesis through distinct signaling networks (Figure 3) [22,28].

### 3.1. PI3K/AKT/mTOR Pathway

Alterations in the phosphatidylinositol 3-kinase/protein kinase B/mammalian target of rapamycin (PI3K/AKT/mTOR) signaling cascade are frequently observed in ECs, especially those with endometrioid features or classified as NSMP. This pathway plays key roles in regulating cellular proliferation, metabolism, angiogenesis, and survival [22,56].

Multi-omics analyses conducted by TCGA [22] and the Clinical Proteomic Tumor Analysis Consortium (CPTAC) [23] have shown that the PI3K/AKT/mTOR pathway dysregulation is frequently found in the NSMP and POLEmut subtypes. Recurrent mutations are often detected in genes including *PTEN*, *PIK3CA*, *PIK3R1*, *CTNNB1*, and *AKT* [57]. Loss of *PTEN* expression is especially common in low-grade EECs [22,23,37]. 

The PI3K/AKT/mTOR pathway is initiated when growth factors such as epidermal growth factor (EGF), insulin-like growth factor (IGF), or certain cytokines bind to receptor tyrosine kinases (RTKs). PTEN, a tumor suppressor, downregulates this process by converting phosphatidylinositol (3,4,5)-trisphosphate (PIP3) back into phosphatidylinositol 4,5-bisphosphate (PIP2). Loss of PTEN function, therefore, results in excess PIP3 accumulation, which in turn drives excessive *AKT* activation and downstream signaling. Mutations in *PIK3CA* or *PIK3R1* further increase the catalytic activity of PI3K, bypassing PTEN control and leading to excessive PIP3 generation. This hyperactivation promotes enhanced AKT phosphorylation, ultimately favoring cell survival, proliferation, and metabolic reprogramming [58]. Although less common, alterations in *mTOR* or the *TSC2* gene can also heighten mTOR complex activity, thereby stimulating cell growth and protein synthesis (Figure 4) [47].

Activation of the PI3K/AKT/mTOR pathway is most prevalent in hormone receptor–positive, low-grade tumors, which typically have a favorable prognosis and largely belong to the NSMP subtype [6,21,22,59]. PTEN mutations are detected in about 40–50% of cases, and PIK3CA mutations occur in approximately 30–40 [22,23]. All in all, this pathway is altered in more than half of EC patients. These aberrations activate downstream signaling and provide a rationale for PI3K or mTOR inhibitors as therapeutic options [21,27,60]. Nevertheless, in advanced or recurrent disease, dysregulation of this pathway poses a major therapeutic challenge. It can serve as a bypass route causing resistance to hormone-based treatments such as progestins [6,21]. When the primary hormone receptor signaling is blocked, the hyperactive PI3K pathway can sustain cancer cell expansion and survival by delivering alternative signals [59]. As a result, targeting this pathway has become an appealing strategy, particularly for tumors that have developed hormone resistance.

The effect of various targeted therapy agents has been investigated in ECs. *mTOR* inhibitors such as everolimus and temsirolimus and selective *PI3K* inhibitors are among them. In the GOG-229E trial, the mTOR inhibitor temsirolimus revealed only modest antitumor activity [61]. More promising results were observed in a Phase II study (NCT01068249), where everolimus combined with letrozole provided meaningful clinical benefit in patients with hormone receptor–positive, recurrent disease [60]. These findings were subsequently validated in the randomized Phase II GOG-3007 trial (NCT02228681) [59,62]. Ongoing studies are also assessing other next-generation agents, such as the PI3Kα inhibitor alpelisib, in *PIK3CA*-mutated populations, frequently in combination with endocrine therapies or other targeted drugs [63].

### 3.2. Hormone Receptor Signaling: Estrogen Receptor and Progesterone Receptor

Hormone receptor signaling, especially the estrogen pathway, plays an integral role in the development of ECs. This concept historically underpinned Bokhman’s dualistic model, which separated EC into estrogen-dependent (Type I) and estrogen-independent (Type II) categories [33]. Multi-omics approaches have since integrated this paradigm into the molecular classification system, showing that the expression of estrogen receptor (ER) and progesterone receptor (PR) strongly correlates with specific molecular subtypes and serves as a key biomarker for prognosis and therapeutic decision-making [57]. ECs that express ER and PR are generally associated with favorable clinical outcomes, along with characteristics such as low-grade EECs, limited disease burden, and low rates of LVSI [64]. ER expression is retained in about 70–80% of endometrioid EC, while PR expression is reported slightly lower, approximately 60–70% [65,66,67]. Loss of receptor expression is more common in high-grade or non-endometrioid histology and is associated with worse prognosis [68].

Estrogen receptor alpha (ERα) can be functionally divided into upstream regulators, downstream effectors, and co-regulatory proteins. Upstream signals and co-activators increase the transcriptional activity of genes leading to cell proliferation, while downstream pathways regulate processes such as proliferation, apoptosis, and metastasis. These estrogen-driven effects are amplified when progesterone is absent, as progesterone often counterbalances estrogen signaling [69] (Figure 4). This mechanism explains the reason EC risk is higher in women with obesity, chronic anovulation, or prolonged exposure to unopposed estrogen [1]. Beyond its classical interaction with estrogen (E2), ERα also interfaces with multiple oncogenic signaling pathways, including PI3K/AKT/mTOR, WNT/β-catenin, MAPK, and JAK/STAT cascades [69].

The expression of the progesterone receptor (PR) is largely regulated by estrogen receptor (ER) signaling and occurs in two isoforms, PR-A and PR-B. Upon activation, PR attenuates estrogen-mediated cell proliferation by lowering ER expression and influencing downstream molecules such as insulin-like growth factor 1 (IGF1) and WNT4 [69,70].

Subtype-dependent hormone receptor patterns have been revealed through multi-omics studies. ECs classified as NSMP or MMRd frequently maintain ER and PR expression, a feature associated with more favorable clinical outcomes and an increased likelihood of responding to hormonal treatment. By contrast, p53abn tumors—usually high-grade and not driven by estrogen—often display minimal or absent ER/PR expression, underscoring the reduced role of hormonal pathways and their link to poor prognosis [43].

MMRd tumors often retain ER and PR expression, particularly in low-grade cases, reflecting their endometrioid histology. However, despite hormone receptor positivity, progestin therapy has shown limited efficacy in this molecular subtype, especially in advanced or high-grade tumors. Consequently, although endocrine therapy is frequently applied to NSMP tumors [43,59], patients with MMRd tumors are more likely to respond to immune checkpoint blockade, including agents such as dostarlimab [52].

Therapeutically, hormonal agents including progestins and selective estrogen receptor modulators (SERMs) are used in selected patients, especially those with low-grade, ER/PR-positive tumors who are unsuitable for surgery or seek fertility preservation [27]. Current clinical trials are assessing combinations of hormonal therapy with targeted agents or immunotherapy to improve clinical outcomes [60,71].

In summary, ER and PR signaling pathways remain important in terms of the biology and management of ECs. Expression patterns identified through transcriptomic and proteomic analysis offer insight into tumor characteristics and support personalized treatment.

### 3.3. WNT/β-Catenin Signaling Pathway

The WNT/β-catenin pathway is a fundamental signaling network that controls cell proliferation, differentiation, and stem cell homeostasis [72]. Under basal conditions, without pathway activation, the β-catenin destruction complex remains functional. This multi-protein complex—comprising adenomatous polyposis coli (APC), glycogen synthase kinase 3β (GSK3β), AXIN, and casein kinase 1 (CK1)—phosphorylates β-catenin, leading to its proteasomal degradation and preventing downstream transcription of β-catenin-dependent genes.

Upon WNT ligand binding to Frizzled receptors and LRP5/6 co-receptors, the destruction complex is inactivated. β-catenin subsequently accumulates in the cytoplasm, translocates into the nucleus, and interacts with TCF/LEF transcription factors to activate oncogenic targets such as MYC, CCND1, and MMP7 [47,72].

Multi-omics studies have demonstrated that alterations in this pathway are prevalent in EC, particularly mutations in exon 3 of CTNNB1, which block β-catenin phosphorylation and degradation [22,23,28]. These mutations lead to nuclear accumulation of β-catenin and are reported in about 15–25% of endometrioid EC, and up to 40% in some cohorts [73,74,75]. Transcriptomic data have also shown upregulation of β-catenin targets and downregulation of negative regulators like DKK1 and SFRP1 [76]. These alterations are enriched in the NSMP molecular subtype, particularly in low-grade, early-stage endometrioid tumors [77]. Although these tumors often appear histologically indolent, CTNNB1 mutations have been strongly linked to increased recurrence risk [49]. Therefore, CTNNB1 mutation status is a valuable prognostic biomarker within the NSMP group and can guide individualized management.

Directly targeting the WNT/β-catenin pathway remains challenging due to its essential role in normal physiology. Current strategies include indirect inhibition: (1) tankyrase inhibitors (e.g., OM-153), which stabilize AXIN and promote β-catenin degradation [78]; (2) PORCN inhibitors (e.g., WNT974), which prevent secretion of WNT ligands [79]; and (3) small molecules such as CWP232291 that disrupt the nuclear interaction between β-catenin and TCF/LEF transcription factors [79]. Although no drug has yet gained approval for EC, early-phase trials—such as the phase 1 study of WNT974 in WNT-altered tumors—demonstrate the potential of these agents for patients with CTNNB1-mutated NSMP tumors [80].

### 3.4. DNA Damage Repair (DDR) Pathway

The integrity of the DNA damage repair (DDR) system is crucial for genomic stability, and its disruption is a key factor leading to malignancy. Multi-omics studies have identified distinct EC subtypes defined by specific DDR alterations, which have substantial prognostic and therapeutic implications.

The POLEmut subtype (5–15% of ECs) is defined by pathogenic mutations in the exonuclease domain of DNA polymerase epsilon, leading to an exceptionally high tumor mutational burden [37]. Paradoxically, this group has the most favorable prognosis, likely due to strong antitumor immunity triggered by the abundance of neoantigens [38]. These patients often receive de-escalated adjuvant therapy in line with current guidelines [5].

MMRd tumors (25–30% of ECs) lack one or more MMR proteins (MLH1, PMS2, MSH2, or MSH6) and display MSI-H status [37]. These tumors are moderately aggressive but show profound response to immune checkpoint inhibitors, such as pembrolizumab and dostarlimab, with durable responses demonstrated in the KEYNOTE-158 and GARNET trials [15,16,17,30]. Consequently, MMR status serves as both a molecular classifier and a predictive biomarker for immunotherapy [39,40,41,52].

### 3.5. Cell Cycle Dysregulation and TP53 Axis

Cell cycle dysregulation is a defining feature of the p53abn/CN-H subtype, which accounts for 15–20% of ECs [37]. These tumors exhibit widespread copy-number alterations, TP53 mutations, and an aggressive phenotype characterized by early recurrence and high mortality [37,42]. They are frequently associated with serous carcinoma and a subset of high-grade endometrioid tumors [22]. TP53 alterations represent one of the most important genomic events in EC. Mutations in TP53 are detected in about 20–25% of all EC cases, whereas they are present in more than 90% of serous carcinomas [22]. These mutations define the p53abn group in the TCGA classification and are strongly associated with poor prognosis and aggressive clinical course [27,53,54].

Given their poor prognosis, intensive adjuvant therapy is recommended [43]. HER2 amplification, common in uterine serous carcinoma (USC), has emerged as a therapeutic target. Adding trastuzumab to standard chemotherapy significantly improved progression-free survival in a randomized phase II trial [44]. Other agents under investigation include Wee1 kinase inhibitors (e.g., adavosertib) [81] and strategies targeting CCNE1 amplification [82]. Future refinement of this group using co-mutation profiling may enable more personalized therapy.

### 3.6. Chromatin Remodeling

Chromatin is a dynamic structure composed of DNA, histone proteins, and non-histone proteins. Within the nucleus of eukaryotic cells, chromatin condenses nearly 2 m of genomic DNA to only 5 to 10 μm in diameter. The regulation of chromatin structure, by transitioning between transcriptionally accessible ’open’ states (euchromatin) and condensed ’closed’ states (heterochromatin), is fundamental to controlling gene expression [83]. This process, known as chromatin remodeling, is critical for normal cellular function. Consequently, the disruption of chromatin remodeling pathways is a key oncogenic driver in numerous human cancers, including endometrial cancer (EC). The SWI/SNF complex, a central ATP-dependent chromatin remodeler, is one of the most frequently mutated tumor suppressors in oncology [22,84,85].

Multi-omics studies have revealed that inactivating mutations in genes encoding components of this complex, particularly AT-rich interactive domain-containing protein 1A (*ARID1A*) and *SMARCA4*, are recurrent events in endometrial cancer. These alterations are more frequently reported in endometrioid and clear cell histology subtypes [22]. ARID1A mutations are found in about 40% of cases, especially in endometrioid carcinomas [22,86,87]. Mutations in other SWI/SNF complex members, such as SMARCA4 and SMARCB1, are less common, reported in about 5–10% [88,89,90]. These aberrations disrupt chromatin accessibility and transcriptional regulation, and they may have prognostic impact and potential as targets for epigenetic therapy [91,92]. Together, loss of function in these chromatin-regulating genes makes cancer cells rely on alternative pathways for survival, creating a weakness that can be targeted through synthetic lethality [93]. For instance, cancer cells with *ARID1A* mutations become highly dependent on the activity of enhancer of zeste homolog 2 (EZH2), a core component of the opposing PRC2 complex [91]. This can lead to the clinical investigation of EZH2 inhibitors, such as tazemetostat, which have been evaluated in basket trials that included cohorts of endometrial cancer patients with *ARID1A*-mutated tumors [94]. Furthermore, preclinical evidence suggests that deficiencies in the SWI/SNF complex can also confer sensitivity to other targeted agents, including PARP [95] and ataxia telangiectasia and Rad3-related protein kinase (ATR) inhibitors [96], opening new avenues for personalized treatment in this molecularly defined subgroup.

### 3.7. HER2/Fibroblast Growth Factor Receptor (FGFR)

Growth factor receptor pathways are well-established treatment targets in many malignancies. In EC, HER2 and FGFR signaling have gained particular attention. HER2 amplification is an oncogenic driver in about 25–30% of uterine serous carcinomas (USC) [44,97,98,99], a subtype almost universally classified as p53abn, but is rare in endometrioid EC. This alteration carries strong prognostic significance and has become a therapeutic target. Trastuzumab in combination with chemotherapy significantly improved outcomes in randomized phase II trials [98] and more recently, HER2-directed antibody–drug conjugates (ADCs) such as trastuzumab deruxtecan (T-DXd) have demonstrated promising activity in HER2-expressing solid tumors, including EC as shown in the DESTINY-PanTumor02 trial [100,101,102].

FGFR2 mutations are reported in approximately 10–12% of endometrioid EC, where they promote aberrant receptor signaling and tumor progression. Although their prognostic impact remains less clear, FGFR2 alterations represent a potential therapeutic target. Activating FGFR alterations lead to constitutive signaling that drives cell proliferation and survival [22]. The development of potent small-molecule FGFR inhibitors has enabled a tumor-agnostic approach to treatment, where patients are selected based on the presence of FGFR alterations regardless of tumor origin. For example, the NCT04083976 trial [103] is evaluating the pan-FGFR inhibitor erdafitinib across multiple solid tumors, including gynecologic malignancies such as EC. These trials highlight a precision medicine strategy for another molecularly defined subset of patients [104].

## 4. Current Applications of Multi-Omics: From Diagnosis to Therapy

The introduction of multi-omics technologies into the clinical management of ECs represents a paradigm shift—from a one-size-fits-all approach to a new era of precision medicine. The molecular insights detailed in the previous sections are no longer confined to research laboratories. They are now being actively applied in real-world clinical setting. This section will explore the ongoing clinical applications of multi-omics, from initial diagnosis to the management of advanced disease (Table 3). 

### 4.1. Risk Stratfification

One of the most immediate impacts of multi-omics in ECs management is patient risk stratification. Traditionally, a solely histopathologic system has been utilized for risk assessment. This system, however, has significant limitations in accurately predicting clinical outcomes due to the underlying diversity of molecular pathways of ECs. The TCGA-based molecular classification [22] has been adopted by ESGO/ESTRO/ESP [27] and FIGO [5] in 2021 and 2023, respectively. They incorporated molecular classification into the conventional histopathological grading system. This approach provides better prognostic accuracy, enabling individualized treatment.

For instance, ECs with POLEmut are known to have an excellent prognosis, regardless of histologic grade. Therefore, patients with POLEmut subtype are recommended to receive de-escalated therapy [5,27,123]. It protects patients from unnecessary intense treatment and associated toxicities. In contrast, ECs with p53abn show poor prognosis, which justifies more intense therapy to reduce the risk of recurrence. The clinical application of molecular classification for risk assessment has already been validated in many studies [38,43,48,123,124]. In addition, NSMP subtype, which is regarded as an intermediate risk group, can be further stratified, as it exhibits substantial variability in recurrence risk [48]. To be specific, the presence of CTNNB1 exon 3 mutations [49], L1CAM overexpression [50], or progesterone receptor loss [51] in NSMP group are associated with poor outcomes. Patients with these alterations, therefore, are likely to benefit from more intensive surveillance or adjuvant therapy [48,50].

Furthermore, molecular risk assessment can be useful to determine fertility-sparing strategies [27]. Conservative management can be more strongly taken into account for patients with early-stage, low-grade ECs—particularly within the POLEmut or NSMP categories.

### 4.2. Precision Targeting of Pathway

The most promising clinical application of multi-omics in EC is identifying specific molecular alterations to enable targeted therapy. This approach is based on finding mutations or biomarkers that can be used to select therapies to which the tumor is likely to respond.

To date, immune checkpoint inhibitors for MMRd or MSI-H tumors have shown the most successful results. The high neoantigen load in this subtype makes them highly susceptible to anti-programmed cell death protein 1 (anti-PD-1) or anti-programmed death-ligand 1 (anti-PD-L1) therapies [39]. These include pembrolizumab and dostarlimab, which are part of the standard management for advanced or recurrent MMRd EC based on the results of pivotal trials like KEYNOTE-158 and GARNET [40,52].

Meanwhile, in p53abn ECs, human epidermal growth factor receptor 2 (HER2) amplification serves as a key biomarker for HER2-targeted therapies. HER2 amplification is found in 25–30% of USCs, a subtype largely classified as p53abn [22,42]. The addition of the anti-HER2 monoclonal antibody, trastuzumab, to standard chemotherapy was shown to significantly improve progression-free survival in a randomized phase II trial, marking an early success for targeted therapy in this high-risk group [97]. More recently, antibody-drug conjugates (ADCs) such as trastuzumab deruxtecan (T-DXd) have shown remarkable efficacy in heavily pretreated patients with HER2-expressing solid tumors, including EC [100,101].

For the majority of ECs driven by hormone signaling, which are mostly categorized as the NSMP subtype, multi-omics has guided the development of combination strategies to overcome therapeutic resistance. The PI3K/AKT/mTOR pathway is frequently co-activated in these tumors and is a known escape mechanism contributing to resistance against endocrine therapy [59]. To overcome this, clinical trials have shown that dual inhibition of these pathways is effective. The combination of the mTOR inhibitor everolimus and the aromatase inhibitor letrozole demonstrated significant clinical benefit in patients with recurrent, hormone receptor-positive EC [60]. Similarly, combining cyclin-dependent kinase 4/6 (CDK4/6) inhibitors like palbociclib with letrozole has also been investigated as a promising strategy for this patient group [112].

Future precision medicine strategies are increasingly focused on exploiting synthetic lethality—targeting a pathway that becomes essential for survival only when another specific gene is lost. For instance, cancer cells with inactivating mutations in ARID1A become highly dependent on the opposing polycomb repressive complex 2 (PRC2) component, EZH2, creating a therapeutic vulnerability [91]. This has led to basket trials of EZH2 inhibitors in patients with ARID1A-mutated solid tumors, including EC [94]. Furthermore, combining poly (ADP-ribose) polymerase (PARP) and WEE1 inhibitors has shown promising activity in p53abn/CCNE1-amplified serous carcinomas, targeting the specific DNA damage response vulnerabilities created by these alterations [81,116,117,118,119].

### 4.3. Emerging Diagnostic and Monitoring Tools

In addition to risk assessment and precision therapy, multi-omics is driving the development of innovative, less invasive tools for early diagnosis and real-time disease monitoring.

Liquid biopsy, an analysis of circulating tumor DNA (ctDNA) from blood sample, has great potential for the management of EC. In addition to blood, sources of liquid biopsy include saliva, urine, cerebrospinal fluid, uterine aspirates, pleural effusions, and even stool [125]. This approach is being investigated for several applications. First, this non-invasive screening method enables early detection of EC in high-risk women by revealing tumor-specific mutations in blood. This method can be more effective for premenopausal or asymptomatic women for whom no established screening guidelines exists [126,127]. Also, molecular analysis of circulating tumor cells (CTCs) can provide information regarding therapeutic targets such as HER2, hormone receptor, or PD-L1 [127]. Beyond initial diagnosis, liquid biopsies offer a dynamic and less invasive way to monitor patients [128]. For example, quantifying changes in ctDNA levels can assess tumor shrinkage or growth in response to therapy, often more rapidly than conventional imaging [129]. Furthermore, identifying the persistence of ctDNA after surgery may indicate the presence of minimal residual disease, which can predict a high risk of relapse and allow for earlier intervention [130]. For patients who have relapsed, profiling ctDNA can reveal new mutations that confer drug resistance, which helps guide the choice of subsequent therapies [128].

Along with these molecular monitoring techniques, advanced imaging and artificial intelligence (AI) are also emerging as powerful tools. Radiogenomics aims to link imaging features from MRI or PET scans with underlying genomic data, potentially allowing for the non-invasive prediction of a tumor’s molecular subtype [131]. Additionally, AI algorithms are being developed to identify molecular features, such as MMR status or p53 mutations, directly from routine histopathology slides. This could streamline and democratize molecular testing, making it more accessible [132]. These emerging tools promise to make the management of EC more dynamic and proactive, enabling earlier intervention and more personalized surveillance throughout the course of the disease.

## 5. Conclusion and Future Perspectives

Endometrial cancer (EC) remains one of the most common malignancies in women, ranking seventh worldwide [2]. Its pathogenesis has long been associated with unopposed estrogen exposure, obesity, diabetes, and metabolic syndrome, traditionally leading to its classification into two types [4]. Type 1 EC, typically diagnosed in pre- or perimenopausal women, is estrogen-dependent, often associated with obesity and metabolic risk factors, and generally carries a favorable prognosis [1,33]. Type 2 EC, by contrast, is more frequently observed in postmenopausal, often non-obese women, and exhibits more aggressive biological behavior with poorer outcomes [1,33]. While this binary model provided useful clinical guidance for decades, advances in molecular biology have reshaped our understanding of EC, leading to a more refined genomic-based classification. The Cancer Genome Atlas (TCGA) research network has identified four molecular subtypes—POLE ultramutated, microsatellite instability-high (MSI-H) hypermutated, copy-number low/p53-wild type, and copy-number high/p53abn—each with distinct molecular alterations, clinical behavior, and prognostic implications [22]. These findings underscore the biological heterogeneity of EC and the need for personalized therapeutic strategies [1].

The cornerstone of EC management is still surgical staging, which provides critical information for individualized adjuvant treatment planning [5,27]. For patients with early-stage disease who are fit for surgery, treatment typically consists of a radical hysterectomy or simple/modified radical hysterectomy, often accompanied by pelvic and para-aortic lymphadenectomy [27]. In advanced-stage disease, surgery plays a primarily cytoreductive role, aiming to reduce tumor burden, and may also include comprehensive nodal dissection [5]. Although there are no randomized clinical trials directly comparing surgical approaches, retrospective studies suggest that complete surgical staging—including lymphadenectomy—may confer therapeutic and prognostic benefits [1]. Given the complexity of decision-making, particularly regarding the extent of lymphadenectomy and adjuvant therapy, surgical planning should be guided by a multidisciplinary team to minimize both undertreatment and overtreatment [27].

Integration of multi-omics data is necessary to fully understand EC biology and to translate it into clinical practice [133]. Another future direction is the use of artificial intelligence (AI) to support this integration. AI and machine learning models can combine genomic, transcriptomic, proteomic, and metabolomic profiles, and may predict diagnosis or prognosis better than using single markers [133,134,135,136]. Some recent studies already applied machine learning to classify EC molecular subtypes and to predict treatment response, but the results are still at an early stage [132,135,137]. Further validation and larger patient cohorts are required. Nevertheless, AI-based multi-omics integration will be an important tool for precision medicine in EC [133,138].

The molecular heterogeneity of EC has direct implications for clinical practice. Distinct genomic alterations, such as mutations in *PTEN*, *PIK3CA*, and *ARID1A*, along with chromosomal abnormalities, are increasingly recognized as key drivers of disease biology and potential therapeutic targets [22,47]. Advanced technologies, including CRISPR gene editing, single-cell genomics, and spatial transcriptomics, have enhanced our understanding of EC at an unprecedented resolution, while integrative multi-omics approaches combining genomics, transcriptomics, proteomics, and metabolomics offer a comprehensive view of tumor biology [59]. These insights are paving the way for precision oncology, where targeted therapies—such as PI3K/AKT/mTOR pathway inhibitors—are tailored to individual molecular profiles, with the potential to improve treatment efficacy while minimizing toxicity [59]. However, translating these genomic findings into routine clinical care presents significant challenges, including variability in patient response, integration of complex genomic data into clinical workflows, and ethical considerations surrounding genetic testing and personalized treatment [1]. Addressing these challenges requires close collaboration among geneticists, oncologists, pathologists, bioinformaticians, and other specialists [1,59]. As genomic science continues to evolve and therapeutic strategies become more refined, there is growing optimism that future management of EC will become increasingly precise, effective, and patient-centered, ultimately improving both survival and quality of life [59].

## Figures and Tables

**Figure 1 cells-14-01404-f001:**
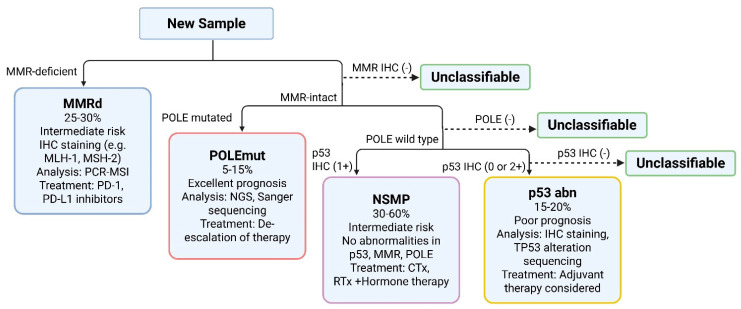
ProMisE algorithm for the molecular classification of endometrial cancer. Adapted from [35] Published by MDPI, Basel, Switzerland, under the terms and conditions of the Creative Commons Attribution (CC BY) license (https://creativecommons.org/licenses/by/4.0/, accessed on 28 August 2025). Abbreviations: MMR, mismatch repair; MMRd mismatch repair deficient; IHC, immunohistochemistry; POLE, DNA polymerase epsilon gene; POLEmut, POLE ultramutated; NSMP, no specific molecular profile; PCR-MSI, polymerase chain reaction-based microsatellite instability; PD-1, programmed death-1; PD-L1, programmed death-ligand 1; NGS, next-generation sequencing; CTx, chemotherapy; RTx, radiotherapy; p53abn, abnormal TP53. Created with BioRender.com (accessed on 28 August 2025).

**Figure 2 cells-14-01404-f002:**
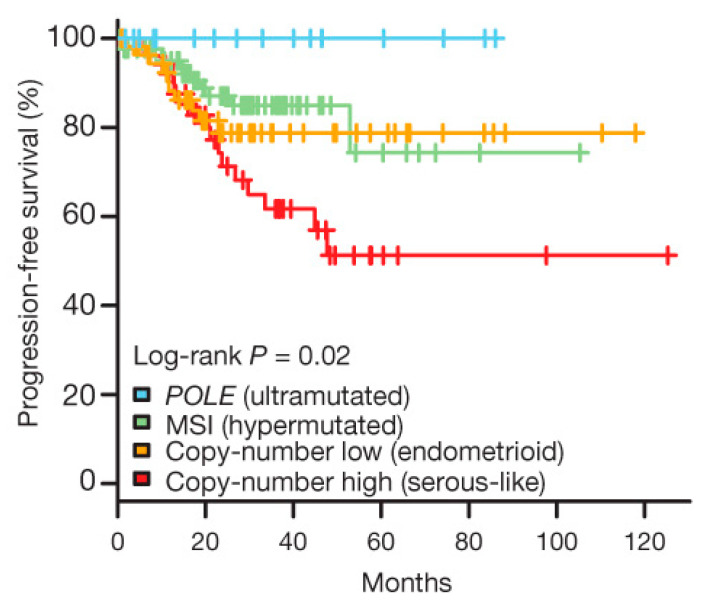
Kaplan–Meier survival curves for progression-free survival across the four TCGA-defined molecular subtypes of ECs. Adapted from [22], distributed under the terms of the Creative Commons Attribution-NonCommercial license (CC BY-NC). Abbreviations: EC, endometrial cancer; TCGA, The Cancer Genome Atlas; POLE, DNA polymerase epsilon; MSI, microsatellite instability.

**Figure 3 cells-14-01404-f003:**
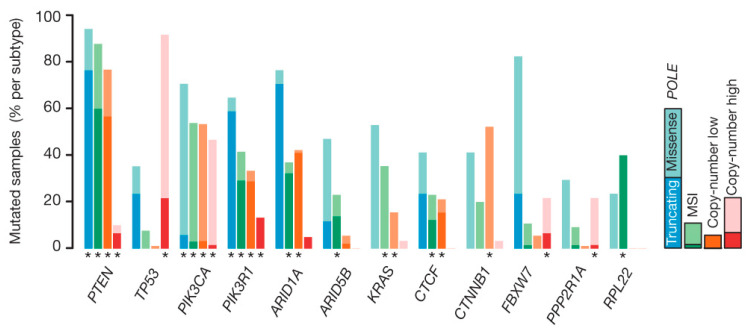
Recurrent alteration of genes among four subgroups in ECs. Adapted from [22], distributed under the terms of the Creative Commons Attribution-NonCommercial license (CC BY-NC). Abbreviations: EC, endometrial cancer; TCGA, The Cancer Genome Atlas; POLE, DNA polymerase epsilon; MSI, microsatellite instability; PTEN, phosphatase and tensin homolog; PIK3CA, phosphatidylinositol-4,5-bisphosphate 3-kinase catalytic subunit alpha; ARID1A, AT-rich interactive domain-containing protein 1A; CTNNB1, catenin beta 1; TP53, tumor protein p53. Asterisk denotes FDR < 0.05.

**Figure 4 cells-14-01404-f004:**
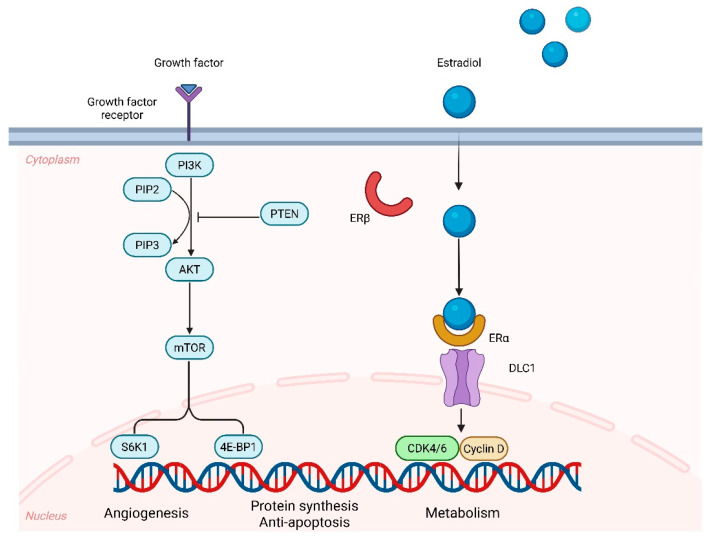
PI3K/AKT/mTOR and estrogen pathway in endometrial cancer. Abbreviations: PI3K, phosphatidylinositol-3-kinase; PIP2, phosphatidylinositol-4,5-bisphosphate; PIP3, phosphatidylinositol-3,4,5-trisphosphate; PTEN, phosphatase and tensin homolog; AKT, protein kinase B; mTOR, mammalian target of rapamycin; S6K1, ribosomal protein S6 kinase beta-1; 4E-BP1, eukaryotic translation initiation factor 4E-binding protein 1; ERα, estrogen receptor alpha; ERβ, estrogen receptor beta; DLC1, dynein light chain 1; CDK4/6, cyclin-dependent kinase 4 and 6. Created with BioRender.com (accessed on 28 August 2025).

**Table 1 cells-14-01404-t001:** Summary of biomarkers in EC.

Gene/Protein	Alteration	Clinical implication
PTEN	Loss-of-function mutation	Early molecular alteration, tumor suppressor [22]
PIK3CA	Activating mutation	Pathway activation, potential therapeutic target [22,23]
TP53	Mutation/overexpression	Poor prognosis, Type II EC [22,27]
ARID1A	Mutation	Chromatin remodeling defect [22]
POLE	Exonuclease domain mutation	Ultra-mutated, good prognosis [22,27]
MSI genes	MSI-H phenotype	Lynch syndrome, prognostic marker [27]
CTNNB1 (β-catenin)	Mutation, nuclear accumulation	Tumor progression, some prognostic value [28,29]
KRAS	Mutation	MAPK pathway activation [22]
Annexin A2, HSPs	Protein overexpression	Candidate circulating biomarkers [30,31,32]

Abbreviations: EC, endometrial cancer; MSI, microsatellite instability; HSPs, heat shock proteins; MAPK, mitogen-activated protein kinase.

**Table 2 cells-14-01404-t002:** Comparison of classification systems in EC.

System	Key Criteria	Subtypes	Prognostic value
Bokhman (1983) [33]	Hormone-dependence	Type 1: estrogen-dependent Type 2: estrogen-independent	Simple model; Limited prognostic accuracy
WHO/FIGO (Histology) [5,27]	Morphology (microscopy), depth of invasion, grade	Endometrioid, serous, clear cell, mucinous, undifferentiated, etc.	Widely used; Inter-observer variability; cannot capture molecular heterogeneity
TCGA (2013) [22]	Molecular profiling	POLEmut, MMRd, p53abn, NSMP	High prognostic accuracy; guides targeted therapy and immunotherapy; Limited by cost and availability

Abbreviations: EC, endometrial cancer; WHO, World Health Organization; FIGO, International Federation of Gynecology and Obstetrics; TCGA, The Cancer Genome Atlas; POLEmut, POLE ultramutated; MMRd, mismatch repair deficient; NSMP, no specific molecular profile.

**Table 3 cells-14-01404-t003:** Ongoing clinical trials of molecularly targeted therapies potentially applicable to endometrial cancer.

Pathway	Identifier	Drug	Mechanism	Status	Relevance to EC	Reference
PI3K/AKT/mTOR	GOG-229E	Temsirolimus	mTOR inhibitor	Completed	Modest activity in recurrent EC	[105]
PI3K/AKT/mTOR + Hormone	NCT01068249	Everolimus Letrozole	mTOR inhibitor + aromatase inhibitor	Completed	Clinical benefit in HR+ recurrent EC	[106]
PI3K/AKT/mTOR + Hormone	NCT02228681 (GOG-3007)	Everolimus Letrozole	mTOR inhibitor + aromatase inhibitor	Completed	Confirmed benefit in HR+ recurrent EC	[107]
PI3K/AKT/mTOR	NCT04049929	YY-20394 (Linperlisib)	Selectively inhibits PI3Kδ isoform	Unknown	Primarily for follicular lymphoma; potential expansion to other solid tumors with PI3K alterations	[108]
PI3K/AKT/mTOR + Hormone	NCT05082025	Copanlisib Fulvestrant	PI3K inhibitor + estrogen receptor	Active, not recruiting	Direct EC application; combinational strategy for hormone-sensitive tumors	[109]
PI3K/AKT/mTOR	NCT01289041	BKM120 (Buparlisib)	Pan-PI3 inhibitor	Completed	Single-agent trial in advanced EC	[110]
PI3K/AKT/mTOR	NCT02549989	LY3023414	Dual PI3K/mTOR inhibitor	Completed	EC-specific study; relevant to recurrent disease	[111]
Hormone pathway	NCT02730429	Letrozole Palbociclib	Aromatase inhibitor + CDK4/6 inhibitor	Completed	Tests a combination of hormone + CDK4/6i for HR+ metastatic EC	[112]
Hormone pathway	NCT03643510	Fulvestrant Abemaciclib	SERD + CDK4/6 inhibitor	Active, not recruiting	Determines the effectiveness of this combination to recurrent EC	[113]
WNT/β-Catenin	NCT03395080	DKN-01 Paclitaxel	DKK1 neutralizing antibody (WNT antagonist)	Completed	Directly included EC	[114]
WNT/β-catenin	NCT02521844	ETC-1922159 Pembrolizumab	PORCN inhibitor + anti-PD-1	Active, not recruiting	Potential synergy in WNT-activated tumors	[115]
DDR/cell cycle	NCT03668340	Adavosertib	WEE1 inhibitor	Active, not recruiting	Single-agent study in recurrent USC	[116]
DDR/cell cycle	NCT02511795	Adavosertib Olaparib	WEE1 inhibitor + PARP inhibitor	Completed	Use of combination of adavosertib and olaparib in refractory solid tumors, Promising results in USC cohort	[117]
DDR/cell cycle	NCT04197713	Adavosertib Olaparib	WEE1 inhibitor + PARP inhibitor	Active, not recruiting	Treats PARP inhibitor resistance solid tumors, including EC	[118]
DDR/cell cycle	NCT04158336	ZN-c3	WEE1 inhibitor	Unknown	For advanced solid tumors, includes a cohort for USC	[119]
TP53 pathway	NCT06413992	Camrelizumab Fluzoparib	Leverages vulnerabilities created by a dysfunctional TP53 axis	Recruiting	Specifically targeting TP53-mutated recurrent or metastatic EC	[120]
TP53 pathway	NCT06521684	None (observational study)	Directly investigates the biological relationship between TP53 mutation and chromosomal instability	Not yet recruiting	To identify new biomarkers and therapeutic targets by analyzing TP53 axis itself	[121]
TP53 pathway	NCT04159155	Combination of chemotherapy and radiotherapy	Compares standard cytotoxic regimens to find the optimal adjuvant therapy for p53abn ECs	Recruiting	To establish the best standard care for high-risk, p53abn EC	[122]
Chromatin remodeling	NCT04104776	Tulmimetostat	EZH2 inhibitor	Completed	Targeted tumors with *ARID1A* or *SMARCA4* mutations, including ECs	[94]
HER2 pathway	NCT04482309	Trastuzumab Deruxtecan	ADC targeting HER2	Recruiting	Major basket trial showing significant activity in various HER2+ solid tumors, including ECs	[100]
FGFR pathway	NCT04083976	Erdafitinib	Pan-FGFR inhibitor	Active, not recruiting	For tumors with *FGFR* alterations, with a cohort for EC	[103]

EC, endometrial cancer; PI3K/AKT/mTOR, phosphatidylinositol 3-kinase/protein kinase B/mammalian target of rapamycin; HR, hormone receptor; CDK4/6i, cyclin-dependent kinase 4/6 inhibitor; SERD, selective estrogen receptor degrader; WNT/β-catenin, wingless/integrated/β-catenin; *DKK1*, dickkopf-related protein 1; *PORCN*, porcupine o-acyltransferase; *WEE1*, WEE1-like protein kinase; USC, uterine serous carcinoma; PARP, poly (ADP-ribose) polymerase; DDR, DNA damage repair; *TP53*, tumor protein p53 gene; p53, tumor protein p53 protein; EZH2, enhancer of zeste homolog 2; *ARID1A*, AT-rich interaction domain 1A; *SMARCA4*, SWI/SNF-related, matrix-associated, actin-dependent regulator of chromatin subfamily A member 4; HER2, human epidermal growth factor receptor 2; ADC, antibody-drug conjugate; *FGFR*, fibroblast growth factor receptor

## Data Availability

No new data were created or analyzed in this study.

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
