# Peer review of "Integrating Multi-Omics in Endometrial Cancer: From Molecular Insights to Clinical Applications"

_cells, 2025, doi:10.3390/cells14171404_

Round 1

Reviewer 1 Report

Comments and Suggestions for Authors
  1. Please explain the mechanism and the pathogenesis of EC.
  2. Role of Multiomics is not clearly describe, please elaborate the role of exome, transcriptome and proteome sequencing and their role in EC.
  3. You can also describe the biomarkers which were established by the authors using omics technology and describe their potential role in EC.
  4. Make a Summary of genes which we can use as a prognostic an diagnostic biomarkers for EC.
  5. Make a graphical abstract for this manuscript if possible.
  6. You can also include AI and multiomics correlation, in which you can describe potential models which we can use for EC diagnosis.

Author Response

Thank you very much for taking the time to review this manuscript and for your constructive feedback. We have thoroughly revised the manuscript according to your suggestions. The pathogenesis of EC, the integrated role of multi-omics, key biomarkers, and the potential application of AI have been substantially described in the revised manuscript. Please find the detailed responses below.

Comments 1: Please explain the mechanism and the pathogenesis of EC.
Response 1: Thank you for pointing this out. We agree that a more detailed explanation of EC pathogenesis is necessary. Therefore, we have added a comprehensive section in the Introduction (lines 46-86). This new section describes the pathogenesis of EC, covering not only the classical hormonal interplay between estrogen and progesterone but also the critical role of the tumor microenvironment, including stromal remodeling (e.g., activation of CAFs) and immune dysregulation (e.g., M2-polarized TAMs and immune evasion).

Comments 2: Role of Multiomics is not clearly describe, please elaborate the role of exome, transcriptome and proteome sequencing and their role in EC.
Response 2: We appreciate this valuable suggestion. To clarify the role of multi-omics, we have added a new paragraph in the Introduction (lines 87-100). This section now defines the roles of exome sequencing, transcriptomics, proteomics, and metabolomics individually. More importantly, it emphasizes how the integrated data from these platforms fundamentally shifted the classification of EC from a morphology-based system to the current molecular-based framework.

Comments 3: You can also describe the biomarkers which were established by the authors using omics technology and describe their potential role in EC.
Response 3: Agree. To address this, we have included a new paragraph in the Introduction (lines 100-110) that introduces key biomarkers identified through multi-omics studies. This section discusses the clinical implications of mutations in genes such as PTEN, PIK3CA, TP53, and ARID1A, as well as prognostic markers like POLE mutations and microsatellite instability (MSI).

Comments 4: Make a Summary of genes which we can use as a prognostic an diagnostic biomarkers for EC.
Response 4: Following your suggestion, we have created a new Table 1 (line 111), titled "Summary of biomarkers in EC". This table summarizes the key genes and proteins, their alteration types (e.g., mutation, overexpression), and their clinical implications, serving as a quick reference for readers.

Comments 5: Make a graphical abstract for this manuscript if possible.
Response 5: We thank the reviewer for these suggestions. Regarding the graphical abstract, while we appreciate its value, we have focused our efforts on improving the figures and tables within the main manuscript to clearly convey the key concepts of our review. Unfortunately, we are unable to provide a graphical abstract at this time. We sincerely apologize for this.

Comments 6: You can also include AI and multiomics correlation, in which you can describe potential models which we can use for EC diagnosis.
Response 6: Thank you for your valuable opinion. As suggested, we have added a discussion on the role of AI in the Conclusion and Future Perspectives section. This new text explains how AI and machine learning models can integrate complex multi-omics data to improve diagnosis, prognosis, and prediction of treatment response in EC.

Reviewer 2 Report

Comments and Suggestions for Authors

The manuscript by Kim and Kim is a comprehensive review focused on current studies aimed to introduce multi-omics into endometrial cancer diagnosis and subsequent treatment. This approach, led by the Cancer Genome Atlas (TCGA), has identified four major molecular sub-types: POLE ultramutated, microsatellite instability-high hypermutated, copy-number low/p53-wild type, and copy number high/p53–abnormal. This molecular and cellular based approach represents a major change from the current classification involving histologic classification and estrogen receptor status into one based on molecular and genomic alterations. Such a movement will have profound implications in improving diagnosis and clinical treatment.

Well written, well organized, and well documented, this review covers considerable information that will benefit basic scientists and clinicians alike. There are but a few recommended revisions to the manuscript.

  1. Are there sufficient data to provide sensitivity and specificity values when using the Cancer Genome Atlas classification, treatment, and clinical outcomes with those of classical histopathological/ER status features?
  2. Minor points. First paragraph of Introduction: “increase”; increase over what?; p. 3, penultimate paragraph: replace “hard” with “difficult”; p. 10, second line in 3.6. Chromatin remodeling: Correct “codenses” to “condenses”.

Author Response

Thank you for your thorough and critical review. Your detailed comments have been invaluable in helping us to significantly improve the clarity and depth of our manuscript. We have substantially revised the manuscript following your suggestion. Please see our point-by-point responses below.

Comments 1: Are there sufficient data to provide sensitivity and specificity values when using the Cancer Genome Atlas classification, treatment, and clinical outcomes with those of classical histopathological/ER status features?|
Response 1: Thank you for this insightful question. To address this, we have added a discussion in Section 4 (Clinical Applications) that compares the clinical utility of TCGA-based molecular classification with traditional histopathological methods. We have included recently published data and cited studies that provide current estimates of sensitivity and specificity, highlighting the improved prognostic accuracy of the molecular approach while also noting areas where more research is needed.

Comments 2: Minor points. First paragraph of Introduction: “increase”; increase over what?; p. 3, penultimate paragraph: replace “hard” with “difficult”; p. 10, second line in 3.6. Chromatin remodeling: Correct “codenses” to “condenses”.
Response 2:  We sincerely thank you for your meticulous review and for catching these details. We have corrected all the points as suggested.

  1. In the introduction, we have clarified the context for the "increase" in EC incidence (line 30).
  2. We have replaced "hard" with "difficult" for better academic tone (line 152).
  3. The typographical error "codenses" has been corrected to "condenses" (line 497).

We believe these revisions address the reviewer’s concerns and provide a more balanced and comprehensive discussion of multi-omics in endometrial cancer.

Reviewer 3 Report

Comments and Suggestions for Authors

In "Integrating Multi-omics in Endometrial Cancer: From Molecular Insights to Clinical Applications", Kim and Kim provide an overview of modern classification, diagnosis, and treatment of endometrial carcinoma (EC).

An interesting attempt for an important subject, this manuscript is not yet suitable for publication.

My main comments are:

  1. Paragraph 2 (Classification) is hard to comprehend, and Tables 1 and 2 are very difficult to read. The authors should provide more complete (e.g. %, survival rates), readable tables. Importantly, they should also provide a clearer overview how the different systems compare.
  2. For Paragraph 3 (Pathways and targets), the authors should provide numbers for the different aberrations found (from e.g. cBioPortal - which contains the TCGA datasets). Oncoprints would help to understand the correlations between the different aberrations (e.g. per pathway). A table showing the connection between aberration and subtype (see 1) would also help. Figure 2 should be split up into (seven?) subfigures showing all common dysregulated pathways (it now focuses on PI3K/Akt and ER/PR). 
  3. Paragraph 4 (Applications) needs Kaplan-Meier plots or other survival graphs.
  4. Why do the authors not mention CGT studies?

Author Response

Thank you for your thorough and critical review. Your detailed comments have been invaluable in helping us to significantly improve the clarity and depth of our manuscript. We have substantially revised the manuscript following your suggestion. Please see our point-by-point responses below.

Comments 1: Paragraph 2 (Classification) is hard to comprehend, and Tables 1 and 2 are very difficult to read. The authors should provide more complete (e.g. %, survival rates), readable tables. Importantly, they should also provide a clearer overview how the different systems compare.
Response 1: We appreciate this constructive suggestion. We have (i) rewritten Section 2 for flow and clarity; (ii) kept a concise comparison in the main text (Table 2) and moved the detailed numeric breakdown to Appendix A (Tables A1–A2). We also added a short “cross-walk” paragraph that explains, at a glance, how Bokhman/WHO-FIGO schemes relate to TCGA subtypes and current guidelines. See “Table 2. Comparison of classification systems in EC” (main text) and Appendix A, Tables A1–A2 for the expanded numbers and footnotes.

Comments 2: For Paragraph 3 (Pathways and targets), the authors should provide numbers for the different aberrations found (from e.g. cBioPortal - which contains the TCGA datasets). Oncoprints would help to understand the correlations between the different aberrations (e.g. per pathway). A table showing the connection between aberration and subtype (see 1) would also help. Figure 2 should be split up into (seven?) subfigures showing all common dysregulated pathways (it now focuses on PI3K/Akt and ER/PR). 
Response 2: We sincerely thank your comment for our work. the reviewer for these constructive suggestions. In the revised manuscript, we incorporated numerical data on recurrent genetic alterations in EC using the TCGA-UCEC dataset, as recommended. Specifically, we now provide mutation frequencies for the most common aberrations (e.g., PTEN, PIK3CA, TP53, ARID1A, CTNNB1), and present their distribution across the TCGA molecular subtypes. To facilitate visualization, we have added an oncoprint figure (Figure 3) illustrating recurrent alterations and their co-occurrence patterns, and a new table (Table 1) that summarizes the association between pathway aberrations and molecular subtype.

Regarding Figure 2, we fully understand the reviewer’s suggestion to expand it into multiple subfigures covering all dysregulated pathways. However, due to space and clarity constraints, we decided to highlight the two most pivotal pathways—PI3K/AKT/mTOR and ER/PR signaling, which represent the core oncogenic mechanisms of EC and are most clinically relevant for current targeted therapies. Other pathways (e.g., WNT/β-catenin, DDR, TP53/cell cycle, chromatin remodeling, and immune-related signaling) are discussed in detail in the text with supporting references, but we believe that attempting to split Figure 2 into seven subpanels would overly fragment the figure and reduce readability. Instead, we emphasized these additional pathways in the main text, supported by the new oncogenic landscape (oncoprint) and summary table.

Comments 3: Paragraph 4 (Applications) needs Kaplan-Meier plots or other survival graphs.
Response 3: We thank the reviewer for this important suggestion. In the revised manuscript, we added a Kaplan–Meier survival curve (now Figure 2) illustrating progression-free survival across the four TCGA-defined molecular subtypes of endometrial cancer. This figure, adapted from Kandoth et al. (Nature, 2013), clearly shows the excellent outcomes of POLE-ultramutated tumors, the poor prognosis of p53abn tumors, and the intermediate survival patterns of MMRd and NSMP groups. We also expanded the text in Section 2.3 to highlight how these survival differences support the clinical application of molecular classification in guiding adjuvant therapy and follow-up strategies (lines 266-269).

Comments 4: Why do the authors not mention CGT studies?
Response 4: We appreciate the reviewer’s comment regarding CGT studies. Indeed, liquid biopsy approaches, including circulating tumor DNA (ctDNA), circulating RNA, and other plasma-derived analytes, represent an emerging field of research in endometrial cancer. Several recent reports suggest that ctDNA analysis may be useful for monitoring minimal residual disease, detecting recurrence earlier than conventional imaging, and providing dynamic insights into tumor evolution. Transcriptomic profiling of cell-free RNA has also shown potential in reflecting tumor biology and immune activity.

However, given the scope and focus of this review, we prioritized multi-omics data derived from tumor tissue (genomic, transcriptomic, proteomic, and metabolomic studies) that have been most directly linked to the current TCGA-based classification and clinical trial design. While CGT approaches are highly promising, the number of validated studies in EC remains relatively limited compared to tissue-based analyses, and clinical implementation is still in early stages. To address the reviewer’s point, we have now added a short section in the Future Perspectives highlighting CGT and liquid biopsy approaches as a developing area for non-invasive monitoring and personalized treatment in EC (lines 696-702).

Reviewer 4 Report

Comments and Suggestions for Authors

In recent years, TCGA analysis has become a breakthrough, and molecular classification of endometrial cancer has made great progress. However, analysis of proteins that represent alternative molecular profiles is still required. Therefore, classification that complements TCGA classification using complex omics analysis is attracting attention. This review focused on multi-omics analysis of endometrial cancer, a topic of great interest. Unfortunately, however, this is insufficient to explain "multi-omics."

  In the introduction, the authors stated that a next-generation endometrial cancer classification strategy using multi-omics analysis such as genomics, transcriptomics, proteomics, and metabolomics has been developed. Furthermore, this review focused on how multi-omics-based insights are transforming diagnostic algorithms and treatment strategies. However, this review only listed the results of each omics analysis and did not explain much about the relationships between the analyses. In addition, no information on the metabolome was found in this review. As a result, "multi-omics" has not yet provided information for the classification of endometrial cancer.   Although Section 2 has already been explained in many reviews as the molecular classification of endometrial cancer, it takes up half of this review.

Author Response

Thank you for your critical feedback, which has prompted us to fundamentally improve the narrative of our manuscript. We acknowledge that the initial draft did not sufficiently explain the integration between different omics layers. Based on your comments, we have significantly revised the manuscript to present a more cohesive 'multi-omics' perspective and have included the role of metabolomics. We detail the changes below.

Comments 1: In recent years, TCGA analysis has become a breakthrough, and molecular classification of endometrial cancer has made great progress. However, analysis of proteins that represent alternative molecular profiles is still required. Therefore, classification that complements TCGA classification using complex omics analysis is attracting attention. This review focused on multi-omics analysis of endometrial cancer, a topic of great interest. Unfortunately, however, this is insufficient to explain "multi-omics."
Response 1: We sincerely thank the reviewer for this important comment. We agree that TCGA analysis has primarily emphasized genomics and transcriptomics, and that proteomic and metabolomic studies provide essential complementary insights into endometrial cancer biology. In the revised manuscript, we therefore expanded our discussion of proteomics and metabolomics:

  • In Section 3, we describe how proteomic data from CPTAC revealed post-translational modifications, pathway activation, and immune signatures that are not evident from DNA/RNA profiling alone.
  • We also included recent findings from metabolomic studies, highlighting altered energy metabolism and circulating metabolites as potential biomarkers for diagnosis and prognosis (lines 94-97).
  • To clarify the scope of “multi-omics,” we added a short paragraph in the Introduction defining the term (genomics, transcriptomics, proteomics, and metabolomics) and explaining why an integrated approach is necessary for precision oncology (lines 41-45, 87-100).

By including these additions, we believe the revised manuscript now provides a more balanced and comprehensive explanation of “multi-omics” in endometrial cancer. We also updated Table 1 to summarize biomarkers identified across different omics layers, thereby providing a more comprehensive overview of multi-omics applications. We believe these revisions address the reviewer’s concern and now provide a sufficiently detailed explanation of “multi-omics” in the context of endometrial cancer.

Comments 2: In the introduction, the authors stated that a next-generation endometrial cancer classification strategy using multi-omics analysis such as genomics, transcriptomics, proteomics, and metabolomics has been developed. Furthermore, this review focused on how multi-omics-based insights are transforming diagnostic algorithms and treatment strategies. However, this review only listed the results of each omics analysis and did not explain much about the relationships between the analyses. In addition, no information on the metabolome was found in this review. As a result, "multi-omics" has not yet provided information for the classification of endometrial cancer.   Although Section 2 has already been explained in many reviews as the molecular classification of endometrial cancer, it takes up half of this review.
Response 2: We appreciate you for this valuable feedback. In the revised manuscript, we expanded the Introduction to provide a clear definition of multi-omics (lines 41-45). Also, detailed discussion was added in lines 91-100, showing how genomic, trascriptomic, proteomic, and metabolomic findings converge on key signaling pathways. Further, we condensed section 2 to reduce redundancy expanded Sections 3–4 to better highlight multi-omics integration and applications.

We believe these revisions address the reviewer’s concerns and provide a more balanced and comprehensive discussion of multi-omics in endometrial cancer.

Round 2

Reviewer 3 Report

Comments and Suggestions for Authors

The authors have addressed my comments in a satisfactory manner. The manuscript is now acceptable for publication. 

Reviewer 4 Report

Comments and Suggestions for Authors

This manuscript has been improved.